# Association of Circulating microRNAs with Coronary Artery Disease and Usefulness for Reclassification of Healthy Individuals: The REGICOR Study

**DOI:** 10.3390/jcm9051402

**Published:** 2020-05-09

**Authors:** Irene R. Dégano, Anna Camps-Vilaró, Isaac Subirana, Nadia García-Mateo, Pilar Cidad, Dani Muñoz-Aguayo, Eulàlia Puigdecanet, Lara Nonell, Joan Vila, Felipe M. Crepaldi, David de Gonzalo-Calvo, Vicenta Llorente-Cortés, María Teresa Pérez-García, Roberto Elosua, Montserrat Fitó, Jaume Marrugat

**Affiliations:** 1REGICOR Study Group, IMIM (Hospital del Mar Medical Research Institute), 08003 Barcelona, Spain; acamps@imim.es (A.C.-V.); isubirana@imim.es (I.S.); jvila@imim.es (J.V.); fmcrepaldi@hotmail.com (F.M.C.); 2CIBER of Cardiovascular Diseases (CIBERCV), Instituto de Salud Carlos III (ISCIII), 28029 Madrid, Spain; david.degonzalo@gmail.com (D.d.G.-C.); CLlorente@santpau.cat (V.L.-C.); relosua@imim.es (R.E.); 3Faculty of Medicine, University of Vic-Central University of Catalonia (UVic-UCC), 08500 Vic, Spain; eulalia.puigdecanet@umedicina.cat; 4CIBER Epidemiology and Public Health, ISCIII, 28029 Madrid, Spain; 5Departamento de Bioquímica y Biología Molecular y Fisiología and Instituto de Biología y Genética Molecular (IBGM), Universidad de Valladolid and Consejo Superior de Investigaciones Científicas (CSIC), 47003 Valladolid, Spain; nadia.nanuk@gmail.com (N.G.-M.); pcidad@med.uva.es (P.C.); tperez@ibgm.uva.es (M.T.P.-G.); 6Cardiovascular Risk and nutrition Group, IMIM, 08003 Barcelona, Spain; DMunoz@imim.es (D.M.-A.); mfito@imim.es (M.F.); 7CIBER of Obesity and Nutrition (CIBEROBN), ISCIII, 28029 Madrid, Spain; 8MARGenomics, IMIM, 08003 Barcelona, Spain; lnonell@imim.es; 9Biomedical Research Institute SanPau (IIB Sant Pau), 08041 Barcelona, Spain; 10Institute of Biomedical Research of Barcelona (IIBB)–Spanish National Research Council (CSIC), 08036 Barcelona, Spain; 11Cardiovascular Epidemiology and Genetics Group, IMIM, 08003 Barcelona, Spain

**Keywords:** oxidized low-density lipoproteins, microRNAs, myocardial infarction, coronary artery disease, circulating biomarkers

## Abstract

Risk prediction tools cannot identify most individuals at high coronary artery disease (CAD) risk. Oxidized low-density lipoproteins (oxLDLs) and microRNAs are actively involved in atherosclerosis. Our aim was to examine the association of CAD and oxLDLs-induced microRNAs, and to assess the microRNAs predictive capacity of future CAD events. Human endothelial and vascular smooth muscle cells were treated with oxidized/native low-density lipoproteins, and microRNA expression was analyzed. Differentially expressed and CAD-related miRNAs were examined in serum samples from (1) a case-control study with 476 myocardial infarction (MI) patients and 487 controls, and (2) a case-cohort study with 105 incident CAD cases and 455 randomly-selected cohort participants. MicroRNA expression was analyzed with custom OpenArray plates, log rank tests and Cox regression models. Twenty-one microRNAs, two previously undescribed (hsa-miR-193b-5p and hsa-miR-1229-5p), were up- or down-regulated upon cell treatment with oxLDLs. One of the 21, hsa-miR-122-5p, was also upregulated in MI cases (fold change = 4.85). Of the 28 CAD-related microRNAs tested, 11 were upregulated in MI cases-1 previously undescribed (hsa-miR-16-5p)-, and 1/11 was also associated with CAD incidence (adjusted hazard ratio = 0.55 (0.35–0.88)) and improved CAD risk reclassification, hsa-miR-143-3p. We identified 2 novel microRNAs modulated by oxLDLs in endothelial cells, 1 novel microRNA upregulated in AMI cases compared to controls, and one circulating microRNA that improved CAD risk classification.

## 1. Introduction

Although coronary artery disease (CAD) incidence is decreasing in many regions [1,2,3], the number of first events in low-risk population is still high [4]. In the general population without cardiovascular diseases (CVD), prevention strategies at the individual level depend on the predicted CVD risk. CVD risk is calculated with validated risk functions, and is used as a criterion to modulate the intensity of the interventions [5,6]. However, the discrimination capacity of most validated risk functions ranges between 70%–88% [7,8,9,10,11,12]. This fact, along with the high proportion of individuals at low/moderate risk, results in >50% of CAD events occurring in this population, who do not qualify for intensive prevention strategies [4]. An improvement of risk prediction in primary CVD prevention should decrease CAD incidence. However, risk prediction variables have seldom changed in the last decades. Attempts to improve risk functions have mainly focused on including genetic and/or circulating biomarkers, with modest impact on their discrimination and calibration capacity so far [5]. It is, however, expected that discovery of new biomarkers should improve risk stratification, and boost the identification of new biological pathways and therapeutic targets.

MicroRNAs (miRNAs) control essential processes in atherosclerosis, and also after an acute myocardial infarction (AMI) [13]. While there are several studies that have analyzed miRNA expression in CAD patients and controls [14,15,16,17], only two studies have examined miRNAs that improve discrimination of high risk individuals in general population [18,19]. These studies examined a panel of miRNAs without taking into account the role of low-density lipoproteins (LDLs), particularly in their oxidized status (oxLDLs), a causal determinant of atherosclerosis and CAD [20]. OxLDLs induce the expression of adhesion molecules; mediate the inflammatory response, and cause dysfunction and apoptosis of endothelial cells (ECs). OxLDLs also regulate the generation of reactive oxygen species and are involved in vascular smooth muscle cell (VSMC) proliferation and apoptosis [21].

In this study we had three aims: (1) to identify miRNAs differentially expressed in vitro in human ECs and VSMCs, after exposure to ox-LDLs; (2) to analyze whether the miRNAs associated to ox-LDLs exposure, and others previously reported to be associated to CAD, were differentially expressed in AMI cases compared to healthy controls; (3) to determine whether any of the miRNAs identified in objectives 1 and 2 could aid in predicting 10-year incident CAD in general population.

## 2. Materials and Methods

### 2.1. Human Arterial Cell Study

#### 2.1.1. EC and VSMC Isolation

Samples of renal arteries from 3 individuals without CVD were obtained from the COLMAH collection (https://www.redheracles.net/plataformas/en_coleccion-muestras-arteriales-humanas.html). Vessels were divided in pieces within the first 24 h after extraction, and placed in Dulbecco’s modified Eagle’s medium (DMEM).

For EC culture, arteries were longitudinally opened, washed with phosphate-buffered saline (PBS) and incubated for 20–25 min with 1% collagenase type I in their endothelial side. ECs present in the collagenase solution were collected and washed with M199 medium containing 20% of fetal bovine serum (FBS). ECs were subsequently grown in 2 µg/mL fibronectin (Sigma, St. Louis, Missouri, United States) coated dishes, with human EC-specific medium supplemented with EC EGM-2 growth factors (Lonza, Basel, Switzerland), and kept at 37 °C and 5% CO_2_. ECs were identified by their cobblestone morphology.

For VSMC culture, cells were isolated from the medial layer of the vessel after removal of ECs and the adventitia layer. The medial layer was cut in 1–3 mm pieces that were seeded in 35 mm Petri dishes treated with 2% gelatin (Type B from bovine skin, Sigma) in DMEM with 20% FBS, penicillin-streptomycin (100 U/mL each), 5 μg/mL fungizone, and 2 mM L-glutamine (Lonza), and kept at 37 °C and 5% CO_2_. Confluent VSMCs were trypsinized, seeded at 1/3 density, and cultured in control medium (DMEM with 5% FBS, penicillin-streptomycin, fungizone, L-glutamine 5 μg/mL insulin, 2 ng/mL bFGF, and 5 ng/mL epidermal growth factor).

#### 2.1.2. LDL Oxidation and Cell Treatment

Cells were incubated at passages 4–8 with 50 µg/mL of native LDL (nLDL), moderately oxidized LDL (moxLDL), or highly oxidized (hoxLDL) for 24 h, and collected with Qiazol Lysis Reagent for RNA extraction. nLDL was obtained by diluting commercial human LDL (Prospec Pro-562, Rehovot, Israel) to 500 µg/mL in PBS. To obtain moxLDL and hoxLDL, nLDL was incubated with 5 µM CuSO_4_ for 30 min, or with 10 µM CuSO4 for 24 h, respectively. The LDL oxidation reaction was ended adding 200 µM EDTA and then dialyzed with the Slide-A-Lyzer Dialysis Cassete 3.5 MWCO (ThermoFisher, Waltham, Massachusetts, United States). The degree of LDL oxidation was assessed by measuring conjugated diene formation at 234 nm.

### 2.2. Case-control and Case-cohort Studies

Five hundred AMI cases and 500 healthy controls were randomly selected and age- and sex- matched. AMI cases were selected from the patients of the REgistre GIroní del COr/Girona Heart Registry (REGICOR) AMI Registry [22,23] admitted consecutively with first AMI in the reference hospital. Controls were selected from the participants free of CAD symptoms during a 10-year follow-up of the REGICOR population cohort recruited in Girona in 2005 [24]. The methods to obtain baseline data on CV risk factors for the REGICOR population cohort and for the REGICOR AMI Registry, have been detailed elsewhere [22,23,24].

In the case-cohort, we included all 117 10-year incident CAD events from the REGICOR population cohort of 2005, and a random subsample of the cohort at ratio 1:4, paired with cases by age and sex (*n* = 485).

#### 2.2.1. CV Risk Factor Data and Blood Sample Collection

Collected/measured variables included age, sex, height, weight, blood pressure, fasting lipid levels (total cholesterol, triglycerides, and high- and low-density lipoprotein cholesterol (HDL-c, and LDL-c, respectively)), fasting glycaemia, and diagnosis and treatment of hypertension, hypercholesterolemia, and diabetes. Blood samples were collected after a 10–14 h fasting and stored at −80 °C. In the REGICOR AMI Registry, the same variables were obtained by trained personnel from clinical records. Blood samples were collected upon arrival to hospital and stored at −80 °C.

#### 2.2.2. Case-finding Procedures

Participants were classified as CAD incident cases if they had a discharge record suggestive of fatal or nonfatal AMI or angina (ICD-9 codes: 410, 411.0, 411.1, 412, 414, 429; and ICD-10 codes: I21–I25, including subtypes). Death certificates with codes 410–414 (ICD-9) or I20–I22, I24, and I25 (ICD-10) were selected for review of medical records and autopsy results. Events were classified by an expert committee following standardized guideline criteria [25]. Angina was defined according to the presence of symptoms and objective demonstration of ischemia or presence of coronary stenosis.

### 2.3. RNA Extraction and Quality Control

Total RNA from LDL-treated cells was isolated with miRNase Mini kit (Qiagen, Hilden, Germany). RNA extraction from serum samples, of the case-control and case-cohort studies, was performed with the MagMAX mirVana Total RNA Isolation kit (Thermo Fisher Scientific). RNA spike-in controls ath-miR159a and cel-miR-2 were added during RNA extraction from serum samples.

### 2.4. miRNA Expression

#### 2.4.1. Human Arterial Cell Study

RNA samples underwent polyadenylation and biotin-labelling with the Flash Tag Biotin HSR RNA Labeling Kit (Thermo Fisher Scientific), and were hybridized for 16 h to the GeneChip miRNA 4.0 arrays, in a GeneChip Hybridization Oven 640. Arrays were washed and stained in a GeneChip Fluidics Station 450 and scanned in a GeneChip Scanner 3000 7G.

#### 2.4.2. Case-control and Case-cohort Studies

RNA extracted from serum samples and from LDL-treated cells underwent polyadenylation, adaptor ligation, and reverse transcription with the Taqman advanced miRNA cDNA synthesis kit (Thermo Fisher Scientific). cDNA samples were combined with the miR-Amp PCR master mix and transferred to a custom made OpenArray plates with the AccuFill instrument (Thermo Fisher Scientific). Custom plates included 21 miRNAs identified in the LDL cell treatment study, 22 previously reported to be associated with CAD, 6 used as housekeeping miRNAs in previous studies, and 2 spike-ins (Appendix A). OpenArray plates were run on the QuantStudio 12K Flex instrument (Thermo Fisher Scientific). qPCR amplification curves were examined with the Thermo Fisher Cloud software. Observations with an AmpScore >1.1 and for which an exponential curve was present were included in the analysis.

### 2.5. Statistical Analysis

Baseline characteristics of the participants were summarized by the mean, standard deviation and a t-test for continuous variables, and by frequencies and chi-squared or Fisher exact test for categorical variables. When stated *p*-values were adjusted for multiple comparisons with the Benjamini–Hochberg correction [26]. All statistical analyses were performed with the R software version 3.5.2.

#### 2.5.1. miRNA Expression

Human arterial cell study: Background correction, normalization, and summarization of expression were performed with the robust multichip average (RMA) function of the aroma.affymetrix R package. Linear models were fit by comparing RMA summarized expression between the following cell treatment conditions: moxLDL vs. nLDL, and hoxLDL vs. nLDL. Moderated t-statistics of differential expression were obtained by empirical Bayes moderation of the standard errors using the limma R package. We selected for further analysis miRNAs with a |fold change| ≥ 1.5 and a significant *p*-value (<0.05), in moxLDL vs. nLDL or in hoxLDL vs. nLDL, and with a linear/quadratic trend in miRNA expression between the conditions nLDL, moxLDL, and hoxLDL, in at least 1 cell type, assessed with generalized linear models.

Case-control and case-cohort studies: Missing values were Ct values associated with a non-exponential curve and reactions without Ct value. miRNAs with > 90% of Ct missing values or not allowing for plate effect correction, and individuals with > 95% of Ct missing values were excluded from the analysis. Cts were corrected for plate effects with the ComBat function of the sva R package. Global normalization was performed by subtracting the Ct of each miRNA from the mean Ct of all expressed miRNAs (ΔCt).

In the case-control study, the fold change was calculated to assess the association of each miRNA and case-control status as: 2^^ (−(ΔCt cases – ΔCt controls)^). Missing Ct values were censored at the maximum observed value for each miRNA. Means and *p*-values were computed with Kaplan–Meier and log rank test, respectively. In the case-cohort study, an age- and sex- adjusted Cox regression model was fitted to assess the association of each miRNA and time-to-CAD event, taking into account the case-cohort design. Missing Ct values were removed from the analysis. The independent association of the significant miRNAs was examined with a Cox regression model further adjusted for age, sex, smoking, cholesterol level, blood pressure, diabetes.

#### 2.5.2. Validation of miRNA Expression

Similarity of miRNA expression between RMA summarized expression (microarray) and ΔCt normalized expression values (OpenArray) was assessed for the 51 miRNAs tested in the OpenArray plates using the Spearman correlation coefficient.

#### 2.5.3. miRNA Predictive Capacity

The contribution to the predictive capacity of the significant miRNAs, over classical risk factors, was analyzed with the C-statistic and ROC curves. The increment of the C-statistic was calculated as previously described for case-cohort studies [27]. Reclassification was examined with categorical and continuous Net Reclassification Index (NRI), and with the integrated discrimination improvement (IDI) for the Framingham-REGICOR CAD risk function [11]. For the categorical NRI, the cutoff point was 10%. NRI and IDI confidence intervals were obtained by bootstrapping.

#### 2.5.4. Bioinformatics Analyses

Pathway analysis of case-control differentially expressed miRNA targets: Conserved and effective gene targets were obtained from the TargetScanHuman website release 7.2 by selecting those with a cumulative weighted context +++ score < −0.3 and with an aggregate score ≥ 0.3. Pathway analysis was performed with the hypergeometric test implemented in the ReactomePA R package.

Analysis of hsa-miR-143-3p targets in human ECs and SMCs: Hsa-miR-143-3p targets were downloaded from TargetScanHuman. Genes expressed by ECs and VSMCs were obtained from the Gene Expression Omnibus (GSE17676). Raw data from the 4 samples of each cell type were downloaded, annotated, and normalized as suggested in the limma R package for agilent microarray data. Probes were filtered for controls, and probes with no symbol or low expression. Athero-protective and athero-prone gene sets in ECs and VSMCs were defined by reviewing the literature (Appendix A).

### 2.6. Ethics

The study complies with the Declaration of Helsinki and was approved by the Parc de Salut Mar Ethics Committee (2015/6202/I). Participants signed a written informed consent.

## 3. Results

Characteristics of the study participants are presented in Appendix A.

There were 21 miRNAs with differential expression that showed a linear or quadratic significant trend, in ECs or VSMCs, upon treatment with hoxLDL/moxLDL compared to nLDL (Figure 1, Appendix A). In ECs, hsa-miR-197-3p was four times more expressed in cells treated with nLDL compared to hoxLDL/moxLDL. While in VSMCs, hsa-miR-122-5p was 2.5 times more expressed in cells treated with hoxLDL compared to nLDL.

The higher the RMA summarized expression obtained with the microarray, the lower the ΔCt obtained from the qPCR OpenArray (Appendix A). Correlation was −0.475, *p*-value < 0.001. Five out of 21 miRNAs differentially expressed in the microarray were detected by qPCR (hsa-miR-122-5p, hsa-miR-125a-3p, hsa-miR-193b-3p, hsa-miR-193b-5p, and hsa-miR-1229-5p).

The case-control analysis included 476 cases and 487 controls (Appendix A), and 14 miRNAs. Twelve miRNAs were differentially expressed -and upregulated- in AMI cases compared to controls (Figure 2, Appendix A). Hsa-miR-499a-5p, hsa-miR-16-5p, and hsa-miR-133a-3p, had expressions 118, 10, and 9 times higher, respectively, in AMI cases compared to controls.

Predicted targets for the 12 upregulated miRNAs were significantly associated with 53 pathways (Appendix A). The 10 most significantly associated pathways included tyrosine, Mitogen-activated protein kinase (MAP), toll like receptor, and cytokine signaling (Appendix A).

In the case-cohort, we analyzed 14 miRNAs in 105 CAD cases and 455 individuals from the REGICOR cohort (Appendix A). Only hsa-miR-143-3p was significantly associated with time-to-CAD incident events in the age- and sex-adjusted Cox model (hazard ratio = 0.56 (95%Confidence Interval (CI) 0.38; 0.82), *p*-value = 0.003) (Appendix A). The association of hsa-miR-143-3p with CAD was independent of classical risk factors (hazard ratio = 0.56 (95%CI 0.35; 0.88), *p*-value = 0.013).

The inclusion of the ΔCt of hsa-miR-143-3p or of the three most significant miRNAs in the case-control study, did not improve discrimination over classical risk factors (Figure 3A and Appendix A). However, the inclusion of these miRNAs improved most reclassification measures. Hsa-miR-143-3p improved continuous NRI (46.70% (95%CI 6.50; 88.74), *p*-value = 0.026) and IDI (0.07 (95%CI 0.02; 0.110), *p*-value = 0.007). And the combination of the 4 miRNAs improved IDI (0.30 (95%CI 0.12–0.49), *p*-value = 0.001, respectively) (Figure 3B,D). No improvement was observed for categorical NRI (Figure 3C).

There were 497 predicted targets for hsa-miR-143-3p. Expressed genes in ECs and VSMCs were 21,548 and 23,873, respectively. We found 84 athero-protective and 79 athero-prone genes for ECs, and 38 athero-protective and 46 athero-prone genes for VSMCs. There were 307/497 and 427/497 of hsa-miR-143-3p targets expressed by ECs or by VSMCs, respectively. In addition, 296/497 of hsa-miR-143-3p targets were expressed by both ECs and VSMCs. EC and VSMC hsa-miR-143-3p targets included both athero-protective and athero-prone genes (Appendix A).

## 4. Discussion

### 4.1. Main Findings

We identified 21 miRNAs differentially expressed in EC and VSMCs upon treatment with increasingly oxidized LDLs. The miRNA showing the largest expression change in VSMCs (hsa-miR-122-5p) was also upregulated in serum samples of AMI cases compared to controls. Another 11 miRNAs associated to CAD were upregulated in AMI cases as compared to controls, particularly hsa-miR-499-5p. One of the 12 miRNAs upregulated in AMI cases (hsa-miR-16-5p) had not been previously described. Hsa-miR-143-3p was independently and inversely associated with time-to-CAD incident event. The addition of hsa-miR-143-3p to classical risk factors improved significantly CAD risk classification of population according to their cardiovascular risk, although discrimination remained unchanged.

### 4.2. Comparison with Published Reports

Among the 21 miRNAs differentially expressed with increasing LDL oxidation, hsa-miR-122-5p, hsa-miR-125a-3p, hsa-miR-193b-3p, hsa-miR-197-3p, hsa-miR-4632, and hsa-miR-7107-5p, were previously associated with oxLDL related processes such as lipid metabolism, reactive oxygen species, and inflammation [28,29,30,31,32,33]. On the other hand, hsa-miR-122-5p, hsa-miR-125a-3p, and hsa-miR-193b-3p, hsa-miR-193b-5p and hsa-miR-1229-5p were validated by qPCR. Besides confirming the association of ox-LDL-related processes with hsa-miR-122-5p, hsa-miR-125a-3p, and hsa-miR-193b-3p, we identified two novel miRNAs associated with oxLDL treatment: hsa-miR-193b-5p and hsa-miR-1229-5p.

Hsa-miR-122-5p was not only upregulated in VSMCs upon treatment with oxLDLs, but also in AMI cases compared to controls, as reported in a recent meta-analysis [34]. This result adds up to the involvement of the LDL oxidation pathway in AMI patients with presumable coronary atherosclerotic lesions [35].

In addition to hsa-miR-122-5p, we found 11 other upregulated miRNAs in AMI cases compared to controls (hsa-miR-16-5p, hsa-miR-21-5p, hsa-miR-22-3p, hsa-miR-125b-5p, hsa-miR-133a-3p, hsa-miR-143-3p, hsa-miR-145-5p, hsa-miR-146a-5p, hsa-miR-186-5p, hsa-miR-222-3p, and hsa-miR-499a-5p). In accordance with our results, 11/12 overexpressed miRNAs in the case-control study were identified as significantly upregulated in individuals with AMI or CAD in previous studies [14,15,16,17,34,36]. Hsa-miR-499a-5p was the most upregulated miRNA, with a 118-fold expression change, an increase similar to that reported by two previous studies of AMI patients [14,15]. Interestingly, hsa-miR-16-5p was not previously identified as upregulated in AMI or CAD patients, but was upregulated after vascular injury in a rat model in the presence of hindlimb ischaemia and associated with a negative effect on endothelial repair [37].

### 4.3. Signaling Pathways

The pathways associated to the targets of the case-control differentially expressed miRNAs suggest that the combined action of these 12 miRNAs regulates a subset of the individual miRNA functions, particularly inflammation. Taking into account that the most significant pathways identified in our study have already been associated with AMI [38,39] this group of miRNAs could explain, at least in part, either the molecular response to AMI or the chronic response to sub-endothelial deposits of oxLDL in coronary arteries. These 12 miRNAs could be used to determine the atherosclerotic plaque activity, and become a marker of prognosis in AMI patients.

### 4.4. Hsa-miR-143-3p Findings and Role in the Vascular Wall

Our case-cohort study identified hsa-miR-143-3p as the only miRNA associated with time-to-coronary events independently of classical risk factors. Moreover, adding hsa-miR-143-3p, alone or in combination with the three most differentially expressed miRNAs in the case-control study, to the commonly used risk factors included in the Framingham-REGICOR CAD risk function [4,10], improved reclassification measurements. The Hunt investigators identified 10 miRNAs that improved CAD prediction in healthy population. However, these miRNAs were different from the ones identified in the present study, except for hsa-miR-21-5p, which was associated with AMI in the case-control study [18,19]. This discrepancy in the results is probably due to the different methodology applied: while the Hunt investigators used a panel of 179 serum miRNAs, we focused on the miRNAs associated with ox-LDL treatment from a panel of 2578 miRNAs, and on previously reported miRNAs. Although the results differ, both studies suggest that new biomarkers such as miRNAs could be useful to improve CAD risk prediction in general population.

While hsa-miR-143-3p was upregulated in AMI cases compared to controls, a higher expression in healthy individuals was associated with a reduced risk of having an incident CAD event at 10 years. This result reflects the complex time- and cell type-specificity of miRNA regulation. MiR-143-3p is involved in the regulation of the phenotype and function of EC and VSMCs [40,41], and in the initiation of the inflammatory response in the myocardium [42]. The increased expression of hsa-miR-143-3p in AMI patients is probably a response to the acute event [43] and could mediate the initiation of the inflammatory response that is triggered after AMI. In addition, an increase in hsa-miR-143-3p expression could also stimulate vessel stabilization through the transfer of this miRNA from VSMCs to ECs [44]. On the other hand, miR-143 promotes differentiation and represses proliferation of VSMCs [40,45], and is inversely associated with coronary atherosclerosis extension in patients with suspected stable CAD [46]. These results agree with our observation of a protective effect of hsa-miR-143-3p on the CAD risk in healthy individuals. MiR-143-3p, which is highly expressed in the normal vascular wall, would maintain the differentiated state of VSMCs, while its downregulation would be associated to a proliferating and less differentiated phenotype (also known as synthetic phenotype).

The bioinformatics analysis suggested that hsa-miR-143-3p tightly regulates atherosclerosis related-processes in ECs and SMCs. Athero-prone and athero-protective hsa-miR-143-3p targets were found ECs and VSMCs. ECs targets included genes associated with EC activation (MAPK7, PLPP3, SERPINE1, THBS1), response to flow stimulation (SENP2), proliferation, apoptosis and migration (CTGF, NRG1) [47,48,49,50,51,52,53]. On the other hand, atherosclerosis related hsa-miR-143-3p targets in VSMCs included CACNA1C, COL1A1, and PTGS2, which are associated with VSMC contraction, extracellular matrix synthesis, and proliferation [54,55].

### 4.5. Strengths and Limitations

Our study is the first analyzing the contribution of miRNAs to CAD risk prediction in general population with a large sample size, and the first studying the association of miRNAs, induced by oxLDLs, to AMI cases and to CAD incidence.

Our study has some limitations that should be considered. First, we wanted to analyze the contribution of ox-LDL induced miRNAs, to the miRNA response in AMI, and to the miRNA expression in general population to predict future CAD. However, among the miRNAs associated with oxLDL treatment, only one was dysregulated in AMI cases compared to controls, and none was related to CAD incidence. This result is probably related to the different miRNA response to oxLDLs that occurs in vivo and in vitro, and to the miRNA intracellular traffic dynamics. We palliated this limitation by including miRNAs previously associated to CAD. Second, we could only validate 5 out of 21 oxLDL induced miRNAs in the study with EC and SMCs. This low reproducibility could be associated to the cross-talk of hybridization events that can occur in microarrays. If cross-talk is present, an individual microarray spot may measure the expression of different miRNAs rather than of a single miRNA. Third, hsa-miRNA-143-3p targets in ECs and VSMCs were examined with a bioinformatics analysis, so further confirmation with experimental studies is warranted. Finally, we have to mention the limited statistical power of our case-cohort study due to the reduced number of CAD events, which is characteristic of cohort studies.

## 5. Conclusions

We identified two novel miRNAs associated with the effect of oxLDLs on endothelial cells (hsa-miR-193b-5p and hsa-miR-1229-5p) and one novel miRNA associated with AMI (hsa-miR-16-5p). We also found that hsa-miR-143-3p improves the reclassification capacity of the Framingham-REGICOR CAD risk function for general population.

## Figures and Tables

**Figure 1 jcm-09-01402-f001:**
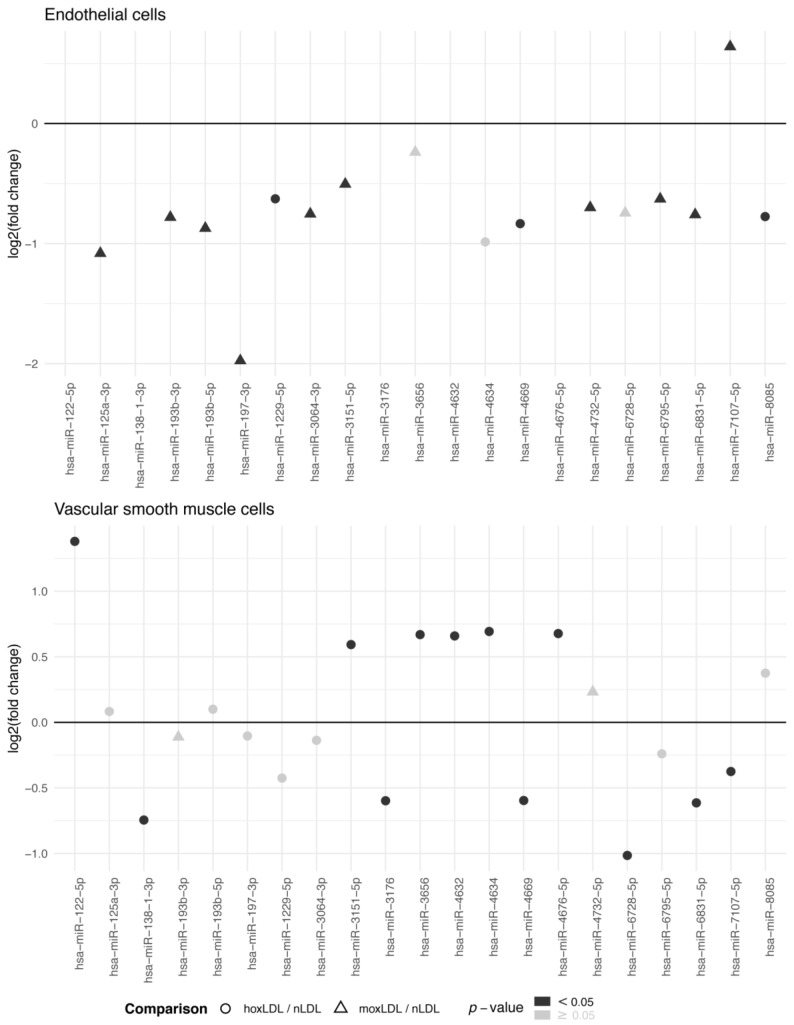
Differentially expressed miRNAs in human endothelial and vascular smooth muscle cells after treatment with oxidized low-density lipoproteins. miRNA expression was measured with microarrays in RNA extracted from cells (obtained from three individuals without cardiovascular disease) treated with native, moderately oxidized or highly oxidized low-density lipoproteins (nLDL, moxLDL, or hoxLDL, respectively). Differential expression of miRNAs was examined with moderated t-tests. Only the differentially expressed miRNAs in at least one cell type which expression followed a linear or quadratic trend between nLDL, moxLDL, and hoxLDL conditions are shown. *p*-values were not adjusted for multiple comparisons. Expression values are presented in Appendix A. Log2 (fold change): binary logarithm of the fold change.

**Figure 2 jcm-09-01402-f002:**
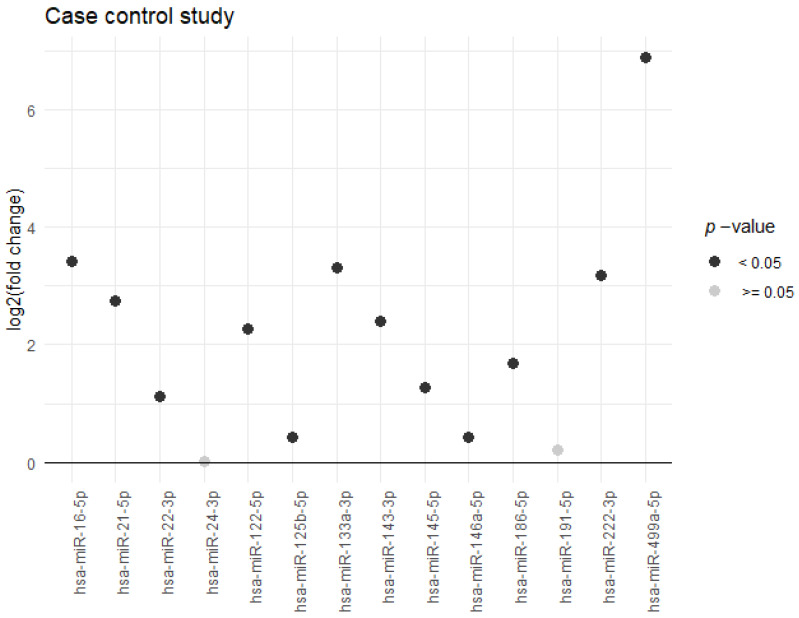
Differentially expressed miRNAs in acute myocardial infarction cases compared to controls. miRNA expression was measured in serum samples of 476 acute myocardial infarction cases and 487 controls, matched by age and sex, using custom OpenArray plates by qPCR. The 14 miRNAs analyzed, with <90% of missing values, are presented. Differential expression of miRNAs was examined with the fold change and log rank test using global normalization. *p*-values were adjusted for multiple comparisons. Expression values are presented in Appendix A. Log2 (fold change): binary logarithm of the fold change.

**Figure 3 jcm-09-01402-f003:**
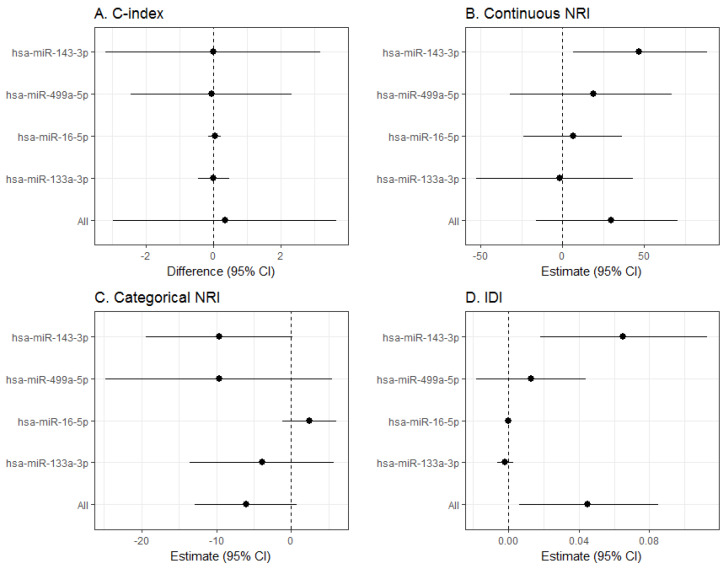
Predictive capacity of the most significant miRNAs identified in the case-control and the case-cohort studies. Change in discrimination and reclassification by including in the Framingham-REGICOR CAD risk function the 3 miRNAs most differentially expressed in the case-control study (hsa-miR-499a-5p, hsa-miR-16-5p and hsa-miR-133a-5p) and the miRNA differentially expressed in the case-cohort study (hsa-miR-143-3p). Discrimination was measured with the C-index (**A**), and reclassification with the net reclassification index (NRI) (continuous (**B**) and categorical (**C**)) and with the integrated discrimination improvement (IDI) (**D**). CI: confidence interval.

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
