# Peer review of "Association of Circulating microRNAs with Coronary Artery Disease and Usefulness for Reclassification of Healthy Individuals: The REGICOR Study"

_jcm, 2020, doi:10.3390/jcm9051402_

Round 1

Reviewer 1 Report

Dégano et al. aimed to examine the association of CAD and oxLDLs-induced microRNAs, and to assess the microRNAs predictive capacity of future CAD events. They identified 2 novel microRNAs modulated by oxLDLs in endothelial cells, and 1 circulating microRNA that improved CAD risk classification.

Although the findings are of potential interest, there are several major limitations and concerns.

  1. The manuscript includes numerous data which make the reader confused. Please summarize the data and convert the manuscript to be more reader friendly.
  2. The author noted that the miRNA improved CAD risk classification, but no ROC curves were provided. (Please have a look at reference [17])
  3. It is unclear why the author needed two types of cohorts.
  4. The manuscript seems to be too long.

Reviewer 2 Report

The authors collected in vitro evidence of miRNA expression changes  from primary endothelial and smooth muscle cells cultures treated with OxLDL. They established a list of responsive miRNAs and added some known miRNAs associated with CAD to perform a screening in serum samples from healthy and CVD patients (MI,CAD). This study merges previously known and new miRNA with potential association to future development of CAD, and includes miR-143-3p in the predicatore of CAD risk.. 

The study is overall well conceived and designed, the data are sound and the all the relevant informations were reported. In addition the authors clearly present and discuss the limits of the study. 

I have some minor recommendations:

  • please state on each table wether you show the adjusted p-value or not.
  • I would suggest to add some more content in the target analysis specific for miR-143-3. The authors should consider adding some in vitro evidence in their system of the miRNA action (e.g. checking for known  target in the specific system-EC and VSMCs-). This would add a relevant information regarding the protective role of this miRNA in CAD development

Round 2

Reviewer 1 Report

The manuscript can be accepted in the current form.

Author Response

We want to thank the reviewer for revising our manuscript again.